# Longitudinal monitoring of SARS-CoV-2 spike protein-specific antibody responses in Lower Austria

Heike Rebholz[1,2,3]☯, Ralf J. Braun[1]☯*, Titas Saha[4], Oliver Harzer[5,6], Miriam Schneider[4], Dennis Ladage[7,8,9]*

**1** Research Division for Neurodegenerative Diseases, Center for Biosciences, Department of Medicine, Faculty of Medicine and Dentistry, Danube Private University, Krems, Austria, **2** Institut de Psychiatrie et Neurosciences de Paris (IPNP), UMR S1266, INSERM, Université de Paris, Paris, France, **3** GHU Psychiatrie et Neurosciences, Paris, France, **4** Department of Scientific Coordination and Management, Faculty of Medicine and Dentistry, Danube Private University, Krems, Austria, **5** Center for Biosciences, Department of Medicine, Faculty of Medicine and Dentistry, Danube Private University, Krems, Austria, **6** Bioscientia, Institute of Medical Diagnostics, Ingelheim, Germany, **7** Internal Medicine, Department of Medicine, Faculty of Medicine and Dentistry, Danube Private University, Krems, Austria, **8** Heart Center, University of Cologne, Cologne, Germany, **9** Department of Pneumology, Maria Hilf Hospital, Mönchengladbach, Germany

☯ These authors contributed equally to this work.
* ralf.braun@dp-uni.ac.at (RJB); dennis.ladage@dp-uni.ac.at (DL)

**Data Availability Statement:** All relevant data are within the paper and its Supporting Information files.

## Abstract

The Lower Austrian Wachau region was an early COVID-19 hotspot of infection. As previously reported, in June 2020, after the first peak of infections, we determined that 8.5% and 9.0% of the participants in Weißenkirchen and surrounding communities in the Wachau region were positive for immunoglobulin G (IgG) and immunoglobulin A (IgA) antibodies against the receptor-binding domain of the spike protein of SARS-CoV-2, respectively. Here, we present novel data obtained eight months later (February 2021) from Weißenkirchen, after the second peak of infection, with 25.0% (138/552) and 23.6% (130/552) of participants that are positive for IgG and IgA, respectively. In participants with previous IgG/IgA positivity (June 2020), we observed a 24% reduction in IgG levels, whereas the IgA levels remained stable in February 2021. This subgroup was further analyzed for SARS-CoV-2 induced T cell activities. Although 76% (34/45) and 76% (34/45) of IgG positive and IgA positive participants, respectively, showed specific T cell activities (upon exposure to SARS-CoV-2 spike protein-derived peptides), those were not significantly correlated with the levels of IgG or IgA. Thus, the analyses of antibodies cannot surrogate the measurement of T cell activities. For a comprehensive view on SARS-CoV-2-triggered immune responses, the measurement of different classes of antibodies should be complemented with the determination of T cell activities.

**Funding:** The author(s) received no specific funding for this work.

**Competing interests:** The authors declare no conflicts of interest.

## Introduction

SARS-CoV-2 infections lead to an adaptive immune response that comprises the participation of virus-specific antibodies of immunoglobulins IgM, IgG, and IgA, as well as cellular responses by various T cell species and other immune cells [1]. IgGs are most prevalent in serum, making up approximately 75% of all serum antibodies and are known to be important at later stages of immunity [1]. IgAs are found in serum as well but can also be secreted in high levels into mucosal surfaces, where they play important roles as first line of defense of the adaptive immune system against viral infections [2, 3].

Many studies on the persistence of antibodies exist for SARS-CoV-2 infections [1–18], but data are still controversial and data informing on the timeline beyond six months are still rare. Nevertheless, positive IgG and IgA results were reported for six months and more [19–21]. Furthermore, prior experience with other human coronavirus species suggests that antibody-based immunity may last for twelve months or more [22].

Various T cell types compose the cellular immune response and are key to protective immunity against COVID-19 [23]. T cells are a heterogeneous group of immune cells, which, among others, can trigger cell death in virus-infected cells. Both CD4+ and CD8+ T cells can develop into memory cells, thereby critically adding to long-lasting immunity. In SARS-CoV-2 infections, the current paradigm states that the T cell-based immune response precedes antibody-based immune responses [24]. Lack of sufficient T cell-based immune responses may cause severe COVID-19 progression [24, 25], and low peripheral T cell counts were associated with non-survival [25], while an early T cell response was linked to a milder form of COVID-19 [26, 27]. T cell response magnitude was significantly higher in patients recovering from severe disease than in patients with mild cases [28]. Thus, SARS-CoV-2-specific T cell activity is critically linked with the clinical course of the disease.

Considering the importance of IgG, IgA, and T cells for the immune response to SARS-CoV-2, we determined 1) spike protein specific IgG and IgA seropositivity, titer levels and persistence and 2) interferon γ release (IGRA) as a readout of T cell activity upon exposure SARS-CoV-2 spike protein-derived peptides in a sample of 552 inhabitants of Weißenkirchen, Austria.

## Materials and methods

Venous blood from 552 participants was collected on the 13th of February 2021 by members of the Lower Austrian Red Cross. Research subjects had to be inhabitants of Weißenkirchen, Austria, to be eligible for enrollment. The cohort consisted of non-infected participants, and formerly infected but recovered cases. We cannot exclude, that cases were also included with asymptomatic infections at the time point of sample collection. Participants were questioned about a prior SARS-CoV-2 infection as well as prior SARS-CoV-2 specific PCR or antigenic tests and their outcomes. Participants were questioned about medical preconditions, previous disease symptoms, including SARS-CoV-2-specific symptoms (*e.g.*, fever, cough and respiratory problems, cold, olfactory dysfunction). Some subjects had already participated at previous data collections [29, 30].

The study was approved by the ethics committee of the Danube Private University, in accordance with local and national guidelines. Participants gave their written informed consent, in the case of minors, informed consent was given by legal guardians. Data were fully anonymized before analysis.

Serum samples were analyzed by in a certified diagnostic laboratory (Bioscientia, Ingelheim, Germany), which participates regularly in inter-laboratory performance testing (ring tests), to ensure the highest accuracy and specificity of the applied tests. An EC-certified semi-quantitative enzyme-linked immunosorbent assay (Euroimmun AG, Lübeck, Germany),

meeting the WHO international standards for anti-SARS-CoV-2 immunoglobulin measurements, was applied for the determination of serum levels of IgG and IgA antibodies, specific for the receptor binding domain of the SARS-CoV-2 (S) spike protein [29]. Serum from a sub-cohort of participants, who had tested positive for IgA or IgG at a previous data collection in June 2020 or presented documentation of a previous SARS-CoV-2 infection, was also used for measurement of T cell activity. Interferon-gamma (IFN-γ) release assay (IGRA) is an *in vitro* blood diagnostic test used in clinical laboratories to measure IFN-γ released by antigen-specific T-cells. In this assay, isolated T cells are incubated overnight with a peptide mix specific to the SARS-CoV-2 spike protein. The release of interferon γ by activated T cells was measured in the very same diagnostic laboratory with an ELISA system (interferon γ release assay, IGRA) according to the protocol of the manufacturer (SARS-CoV-2-IGRA, Euroimmun AG) [31]. The SARS-CoV-2-spike protein-specific activities of subtypes of both the CD4+ and the CD8 + T cell populations can be assessed by this method, although the highest response is from CD4+ subpopulations [32, 33].

For comparisons of Ig titers at two time points, either unpaired (when values of all participants were analyzed) or paired Student's t-tests (for analysis of values from same participants at two times points) were used for statistical analyzes. Curve fit analysis of IGRA as a function of Ig titers was performed with a confidence interval of 95% and automatic outlier elimination (6/70 values were excluded).

## Results and discussion

### Cohort description

In February 2021, 552/1404 (39.3%) inhabitants of Weißenkirchen participated in a voluntary blood draw to have their serum tested. 54.3% of our cohort was female, the mean age was 48.6 years, and the mean BMI was 25.9 (Table 1). Inhabitants were invited to participate in our study through a public call, with no restrictions imposed. 363 subjects had participated in our previous study in June 2020 [29], enabling longitudinal analyses within this subgroup. 50.6% of this subgroup was female, the mean age was 49.2 years, and the mean BMI was 26.1 (Table 1).

**Table 1. Cohort composition.** * Participants with a previous positive test for SARS-CoV-2 (PCR or antigen) were selected for quantification of IGRA.

| | Febr 2021 (all participants) | Participants in June 2020 and Febr 2021 |
|---|---|---|
| **Number of participants** | 552 | 363 |
| **Female** | 300 (54.3%) | 184 (50.6%) |
| **Age (mean +/- SEM)** | 48.6 (+/- 0.8) | 47.4 (+/- 1.4) |
| **BMI (mean +/- SEM)** | 25.9 (+/-0.2) | 25.5 (+/-0.4) |
| **IgG+ (>0.8 AU)** | 138 (25%) | June 2020: 42 (11.6%) |
| | | Febr. 2021: 86 (23.6%) |
| **IgA+ (>0.8 AU)** | 130 (23.6%) | June 2020: 44 (12.1%) |
| | | Febr. 2021: 83 (22.8%) |
| **IgG+ and IgA+ (>0.8 AU)** | 109 (19.7%) | June 2020: 29 (7.9%) |
| | | Febr. 2021: 65 (17.9%) |
| **IGRA SARS2+ (>100 mU/ml)** | 57 of 78 tested* | Febr. 2021: 34 of 50 tested* |

Blood was used for the determination of serum levels of IgG and IgA antibodies, specific for the receptor binding-domain of the SARS-CoV-2 spike protein. For a subgroup of participants with IgG/IgA levels above threshold (N = 78), T cell activity was determined through quantification of interferon γ release (IGRA) by T cells, activated by a SARS-CoV-2-specific peptide mix.

## Prevalence of seroconversion

In our previous study (June 2020), we measured in a different cohort from Weißenkirchen and surrounding communities that 8.5% were IgG positive, 9.0% were IgA positive, and 5.7% were both IgG and IgA positive [29]. In the current study, our measurements for February 2021 indicated that 25.0% (138/552) of participants from Weißenkirchen were positive for IgG values, 23.6% (130/552) of participants were positive for IgA values, and 19.7% (109/552) were positive for both SARS-CoV-2 spike-protein specific IgG and IgA (Table 1). The slight incongruence of percentage of seroconversion in the cohort of June 2020 in Table 1 one compared with the published data [26] stems from the fact that in Table 1 only those participants are represented who donated blood on both dates and more likely were infected. The presented data from June 2020 and February 2021 cannot be directly compared due to different cohort compositions after the first and the second peak of the pandemic. In addition, they cannot easily be extrapolated to the total population of the township, since most probably there was a recruitment bias in our study and people who suspected that they either had been infected with SARS-CoV-2 or had been in contact with COVID-19 patients preferentially enrolled. Nevertheless, our data show that Weißenkirchen was heavily affected by both peaks of the pandemic, but in February 2021 was still far ahead from reaching natural herd immunity, supporting the need for the current extensive vaccination programs.

In February 2021, 10.9% (60/552) of participants stated in the questionnaire having had a prior SARS-CoV-2 positive PCR test, and for 53 (88%) of them, seroconversion could be detected through IgG or IgA positivity. As the incidence of seroconversion of 23.6–25.0% was markedly higher at that time point as predicted from the positive PCR tests, our data hint towards a significantly higher prevalence of SARS-CoV-2 positive people in this Austrian hotspot than officially recorded. This high level of undetected cases is in line with other reports from Austria [34].

## SARS-CoV-2-spike protein-specific IgG and IgA levels and persistence

We previously observed that both IgG and IgA seroprevalence remained very stable in October 2020, *i.e.*, four months after our initial study in June 2020 [30]. In the current study, we were interested in the seroprevalence eight months after our initial study. The 363 participants, who donated blood both in June 2020 and February 2021, allowed us to determine the persistence of serum IgG and IgA antibodies, although the time point of infection or disease onset could not be determined in our cohort.

In June 2020, 11.6% (42/363) of participants had IgG levels above the cutoff (of 0.8 RU), compared to 23.6% (86/363) in February 2021 (Fig 1A). Similarly, in June 2020, 12.1% (44/363) of persons had positive IgA levels, compared to 22.8% (83/363) in February 2021. Thus, the proportion of people with SARS-CoV-2-spike protein-specific antibodies approximately doubled within the eight months between June 2020 and February 2021, supporting our assumption that Weißenkirchen was heavily affected by the second peak of the pandemic.

Next, we plotted the semiquantitative IgG and IgA titers for the 363 donors at both time points. For IgG titers, the mean values of the titers significantly increased from 0.69 RU (+/- 0.08 RU) to 0.96 RU (+/- 0.1 RU) (two-tailed t-test, p = 0.002) (Fig 1B). IgA levels were also higher in February 2021, with 0.74 RU (+/-0.08 RU) compared to 0.48 RU (+/- 0.06 RU) in June 2021 (p < 0.0001) (Fig 1C). These data demonstrate the significant increase of antibody-based immune responses from June 2020 to February 2021 in the subgroup tested at both time points. However, it must be noted that our conclusions about the cohort are limited by the absence of a specific date of disease onset.

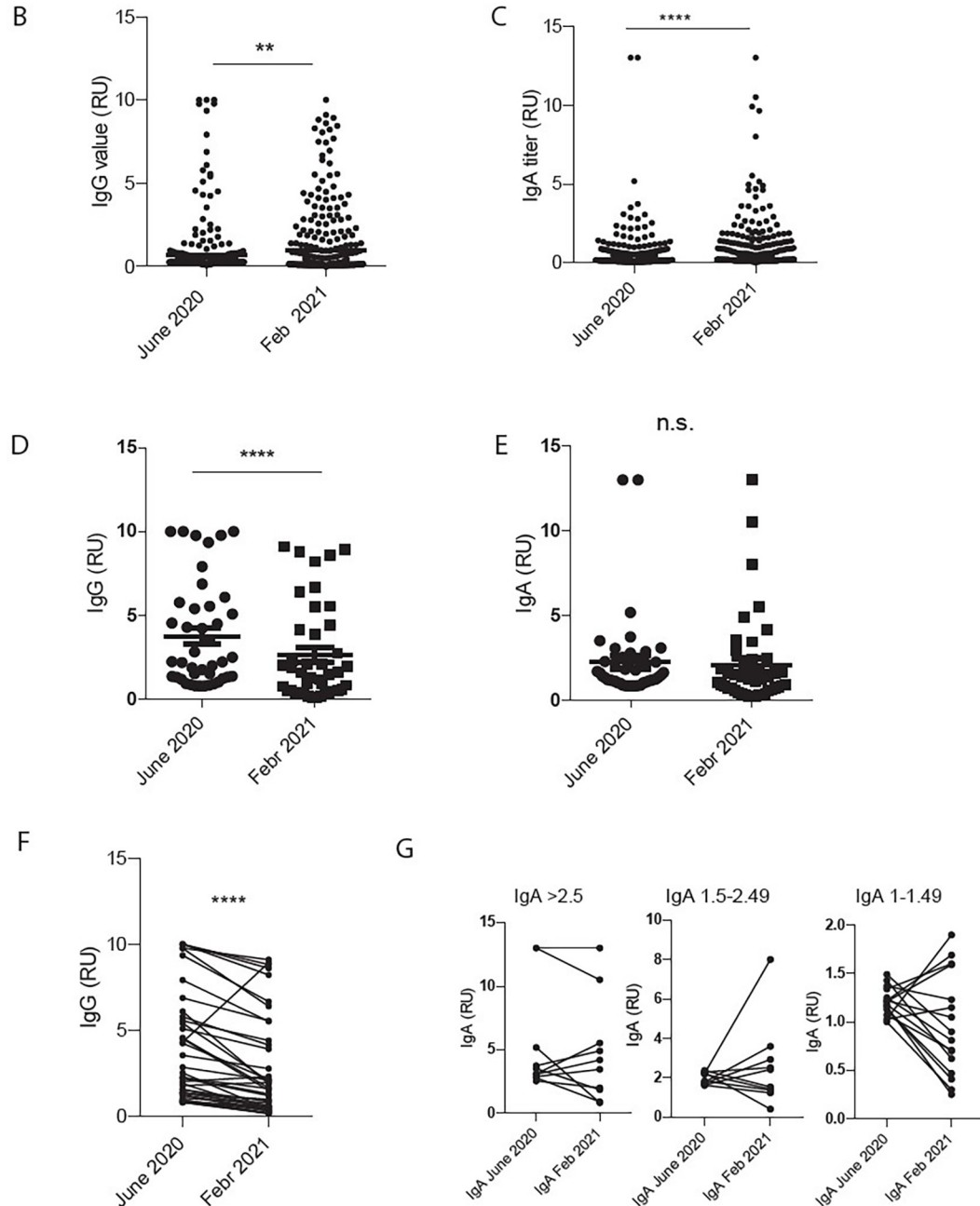

**Fig 1. Immunoglobulin titers measured in June 2020 and February 2021.** Table depicting numbers of participants with above-threshold titers for respective Igs (A). Vertical scatter plots of IgG (B) and IgA (C) titers derived from all participants at two time points, and of IgG (D) and IgA (E) titers from participants who tested positive in June 2020. Plots depicting IgG (F) and IgA (G) titers as in (D,E) but with lines connecting data points of each individual. Statistical analysis was performed using Student's t-test (**p < 0.01, ****p < 0.0001).

Next, we were interested in determining whether this increase in IgG and IgA serum concentration in February 2021 was mainly due to new infections or to a combination of new infections plus stable or increasing antibody titers in SARS-CoV-2 participants from June 2020. Therefore, we restricted the plotting of the Ig values to participants that had been tested positive in June 2020 for IgG (42 participants) and IgA (44 participants), respectively. We found that IgG levels were significantly reduced by 24% from the first measurement in June 2020 to the recent measurement in February 2021 from 5.24 RU (+/- 0.60 RU) to 3.97 RU (+/- 0.57 RU) (p = 0.0007) (Fig 1D). IgA levels remained stable between time points of data collection (Fig 1E). When the same data set is plotted with connecting lines between values from the same individuals, one can observe that almost all IgG values (39/42) are reduced in February 2021 when compared to June 2020 (Fig 1F, p < 0.0001). For IgA values, most values (33/47) are reduced or stable, whereas 30% (14/47) of the values are elevated in comparison to the first measurement in June 2020. To illustrate IgA titer development, we split the data into three bins: IgA levels > 2.5; IgA levels 1.5–2.49; IgA levels 1–1.49 (Fig 1G, from left to right). In most participants with low IgA levels in June 2020, titers decreased between time points, while in participants with moderate to high IgA levels in June 2020, however, titers increased further. One possible explanation for the phenomenon might be an underlying re-infection, which, while rare, has been shown for 0.65% in a large longitudinal population-level observational study [35].

These data indicate that the increase in total IgG and IgA titers from the 363 persons participating in both the June 2020 and February 2021 studies (see Fig 1B and 1C) mainly stems from new infections, although the time points of disease onset could not be identified. However, even though IgG levels of participants that had already seroconverted in June 2020 (or before) are significantly reduced in February 2021, they remain well above the threshold of the applied ELSIA test over the time course of eight months. Taken together, our data show that both IgG and IgA seroprevalence persistent between June 2020 and February 2021.

## The extent of the T cell response does not significantly correlate with either IgG or IgA levels

To ascertain if SARS-CoV-2-spike protein-specific immunoglobulin levels and T cell responses correlate, we tested the blood of a subgroup of 78 participants for T cell activities. T cell activities were determined for participants with IgG/IgA levels above threshold at either time point, by measuring interferon γ release (IGRA) by T cells activated by a SARS-CoV-2-specific peptide mix. Six of the 78 samples (7.7%) showed an IGRA response that was higher than the maximal value of the standard curve and therefore had to be excluded in the plot. From the remaining participants, 64% (50/78) showed SARS-CoV-2-spike protein-specific T cell activation, 58% (45/78) demonstrated SARS-CoV-2-spike protein-specific IgG, 58% (45/78) showed SARS-CoV-2-spike protein-specific IgA levels, and 44% (34/78) showed all three hallmarks of adaptive immunity (Fig 2). The latter group should benefit from a comprehensive immune response, including humoral and cellular adaptive immunity.

When comparing SARS-CoV-2-spike protein-specific T cell activities with SARS-CoV-2-spike protein-specific humoral response, in 86% (43/50) of persons with active T cells, SARS-CoV-2-spike protein-specific IgG levels could be detected (Fig 2). In contrast, only 74% (37/50) of persons with active T cells showed elevated levels SARS-CoV-2-spike protein-specific IgA. To test whether SARS-CoV-2-spike protein-specific T cell activities and levels of IgG and IgA may be correlated, we plotted IgG or IgA titers against the response of the interferon-γ-release assay (IGRA). When we plotted the IgG values of the remaining samples against the corresponding IGRA values, we could not find a significant correlation with the IGRA response (Fig 3A). Moreover, no such correlation was found for IgA values from the same

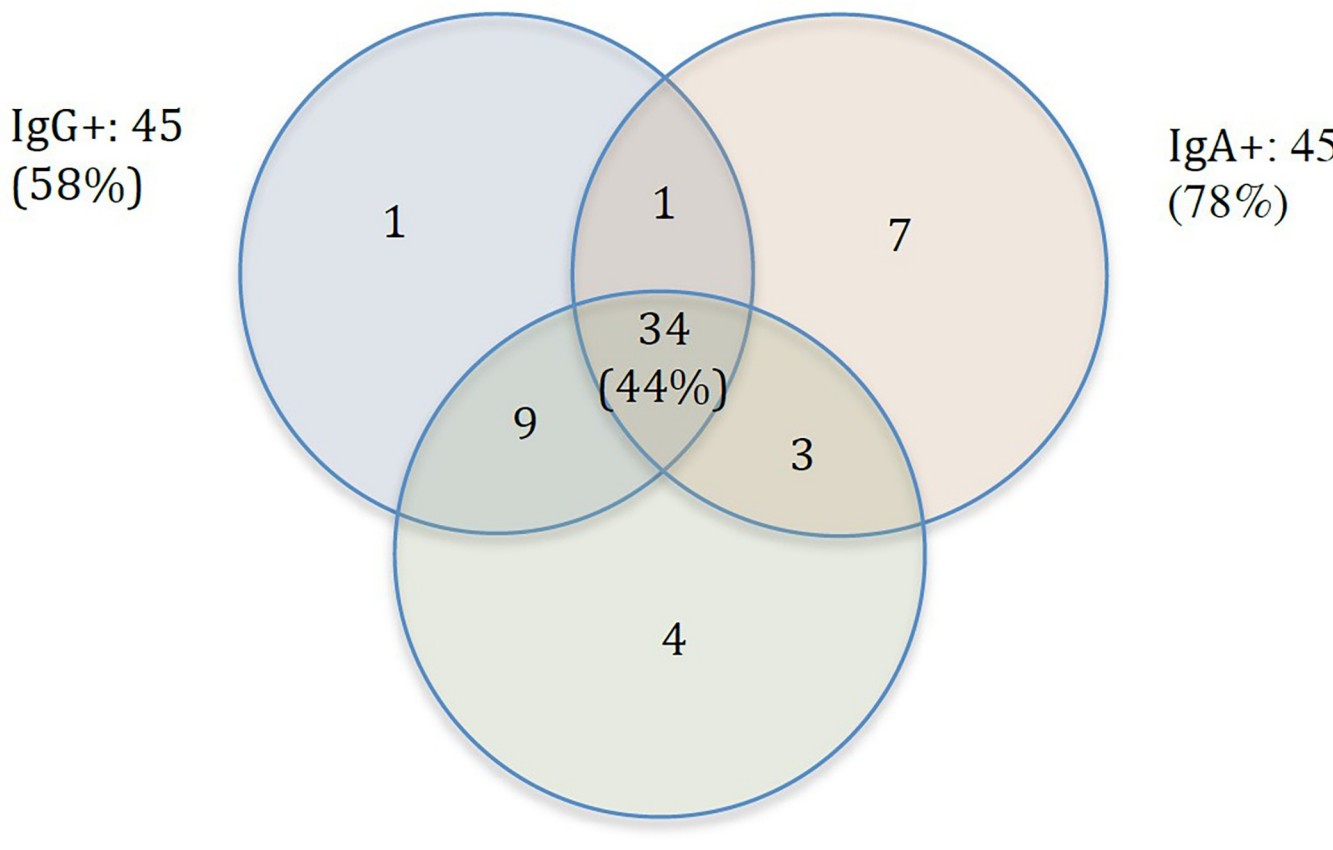

**Fig 2. Venn diagram depicting numbers of participants of the test cohort in February 2021 who had all three parameters (IgA, IgG and IGRA) measured (N = 78).**

blood samples (Fig 3B). Although most participants with SARS-CoV-2-spike protein-specific humoral responses show virus-specific T cell activities, the magnitudes of both responses do not significantly correlate. Thus, for a comprehensive assessment, it is advised to test both the T cell response and the seroprevalence, including IgG and IgA.

## Measuring SARS-CoV-2-spike protein-specific IgG alone does not give a comprehensive view on immune response

One important finding of our study is that IgA titers remain more stable than IgG titers; the latter are supposed to be the carriers of long-term immune response. The literature on

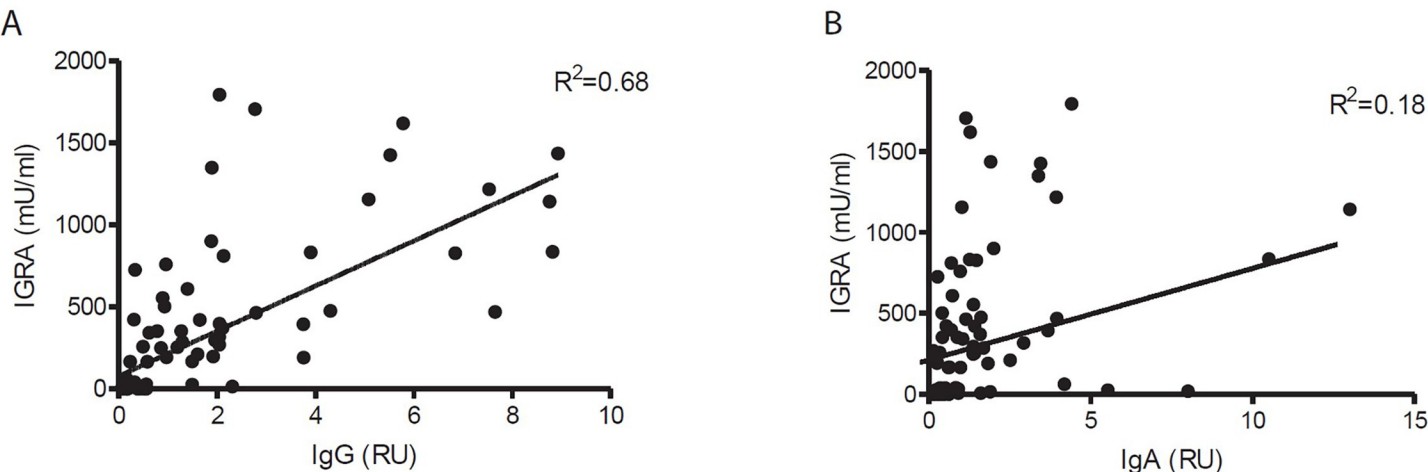

**Fig 3. Correlation of immunoglobulin titers with IGRA response.** Plot of interferon release as a function of IgG (A) and IgA (B) titers derived from serum taken in February 2021.

antibody persistence is ambiguous. For example, one study showed that IgAs against N protein and the receptor binding domain (RBD) of the S protein wane within a period of four weeks [2]. Our findings are consistent with data by Dan et al. who showed that S and RBD IgGs exhibited similar kinetics and were shorter lived than S IgAs. Both antibody types were still largely present in most of their subjects six to eight months after infection [5]. Wang et al. further showed that IgG levels start to decline three months post symptom onset [36]. Differences in the literature can be in part explained by the different antibody targeting specificities, which may also entail different stabilities, but also by the different sensitivities of the applied test systems.

Although most participants of our study with elevated SARS-CoV-2-spike protein-specific antibodies show specific T cell activities, we did not detect a significant correlation between the magnitudes of the humoral and T cell responses at a given time point. Few studies test the same subjects for IgGs, IgAs and the T cell response to ascertain an inter-relationship between the different immune markers. In a relatively small study with twenty subjects with mainly mild COVID-19 symptoms, a clear correlation between RBD IgGs and S-specific CD4+ T cell response was determined, and a similar correlation was found between anti-spike IgA titers and spike-specific CD4+ T cells [37]. In other studies, both mild and severe COVID-19 patients showed SARS-CoV-2-specific-T-cell response, while no difference according to disease severity was observed [38]. Here, we did not detect a correlation between S-specific IgA and IGRA, possibly due to differences in assay technicalities as well as differences in antibody specificities. Another study found a moderate correlation between S-specific IgG antibodies and T cell responses [39]. Dan et al. determined that the ratio between CD4 cell and RBD IgGs is stable over time, however the variation was large, impeding a conclusive statement [5]. They also found higher S-specific IgG values in hospitalized cases, while their T cell memory was reduced. Our data pointing stems solely from non-hospitalized subjects and thus cannot be directly compared with the hospitalized patient group of Dan's study.

The major limitation of our study is the fact that we do not have reliable PCR test results for participants. Since this was a study with no recruitment restriction, participants self-reported previous SARS-CoV-2 infections or COVID-19 symptoms, but no PCR test was performed within our study. Up to 10% of patients with a known infection do not become seropositive. As we restricted T cell analyses to a seropositive subgroup, we may have overseen a certain

number of patients which could have shown a positive IGRA response. Furthermore, since we did not perform PCR tests, it is difficult to clearly determine the maximal period of Ig persistence in our cohort. If patients were seropositive in June, we can deduce antibody persistence of at least eight months (time between June 2020 and February 2021). Especially for patients who were infected during the first wave described in Lower Austria in February 2020, antibody persistence possibly may be extended to twelve months. Indeed, there may be a recruitment bias since people with known or suspected prior infection or prior contact with infected people, may have preferably enrolled in our study.

## Conclusion

In our population study, we show evidence of a high and persistent prevalence of antibodies (both IgG and IgA). Although most participants with antibodies (76%) demonstrate specific T cell activities (and vice versa), we could not find a significant correlation of the extent of T cell activation with the concentration of antibodies. The quantitative determination of both markers of immunity is not useful for routine analyses, due to the increased complexity of T cell measurements. However, the analyses of antibodies cannot surrogate the measurement of T cell activities, which are predicted to markedly contribute to immunity against SARS-CoV-2. As most data sets on immunity after infection or vaccination rely on (IgG) antibody determination only, further studies will have to ascertain to what extent antibody levels can predict immunity. These analyses might be challenged by a broad dispersion within the human population concerning the reciprocal interaction between antibody and T cell activities.

## Supporting information

**S1 Data.**
(XLSX)

## Acknowledgments

We would like to thank district manager Markus Pöschl, managing director Manfred Türk, Dr. Hannes Winkler, Helene Winkler, Dr. Herwig Jamek and Dr. Julia Jamek (all Austrian Red Cross, district office Krems/Donau) and the entire Red Cross team for co-organization and taking the blood samples.

## Author Contributions

**Conceptualization:** Ralf J. Braun, Dennis Ladage.

**Data curation:** Heike Rebholz, Ralf J. Braun, Titas Saha, Oliver Harzer, Miriam Schneider, Dennis Ladage.

**Writing – original draft:** Heike Rebholz, Ralf J. Braun, Dennis Ladage.

**Writing – review & editing:** Heike Rebholz, Ralf J. Braun, Dennis Ladage.

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
