## [Decision Letter · Decision Letter 0]

20 Oct 2021

PONE-D-21-29769

Longitudinal monitoring of SARS-CoV-2-specific immune responses

PLOS ONE

Dear Dr. Braun,

Thank you for submitting your manuscript to PLOS ONE. After careful consideration, we feel that it has merit but does not fully meet PLOS ONE’s publication criteria as it currently stands. Therefore, we invite you to submit a revised version of the manuscript that addresses the points raised during the review process.

An important issue that needs to be clarified is whether the serological test used is calibrated against the WHO international standard for anti-SARS-CoV-2 immunoglobulin. If the test used is not calibrated against the WHO international standard for anti-SARS-CoV-2 immunoglobulin, data may be of little use.

We look forward to receiving your revised manuscript.

Kind regards,

Carlo Torti

Academic Editor

PLOS ONE

Journal Requirements:

2. Please ensure that you have specified (1) whether consent was informed, (2) what type you obtained (for instance, written or verbal, and if verbal, how it was documented and witnessed). If your study included minors, state whether you obtained consent from parents or guardians. If the need for consent was waived by the ethics committee and (3) If you are reporting a retrospective study of medical records or archived samples, please ensure that you have discussed whether all data were fully anonymized before you accessed them and/or whether the IRB or ethics committee waived the requirement for informed consent. If patients provided informed written consent to have data from their medical records used in research, please include this information.

[We would like to thank district manager Markus Pöschl, managing director Manfred Türk, Dr. 

Hannes Winkler, Helene Winkler, Dr. Herwig Jamek and Dr. Julia Jamek (all Austrian Red 

Cross, district office Krems/Donau) and the entire Red Cross team for co-organization and taking 

the blood samples. This study was financed by internal funds of the Danube Private University.]

 [This work was funded by internal funds of the Danube Private University.]

Additional Editor Comments:

Please comply with the Reviewers' comments and consider whether the serological test used is calibrated against the WHO international standard for anti-SARS-CoV-2 immunoglobulin. If the test used is not calibrated against the WHO international standard for anti-SARS-CoV-2 immunoglobulin, data may be of little use to the scientific community.

Reviewers' comments:

Reviewer's Responses to Questions

Comments to the Author

1. Is the manuscript technically sound, and do the data support the conclusions?

Reviewer #1: Partly

Reviewer #2: Yes

2. Has the statistical analysis been performed appropriately and rigorously?

Reviewer #1: Yes

Reviewer #2: Yes

3. Have the authors made all data underlying the findings in their manuscript fully available?

Reviewer #1: Yes

Reviewer #2: Yes

4. Is the manuscript presented in an intelligible fashion and written in standard English?

Reviewer #1: Yes

Reviewer #2: Yes

5. Review Comments to the Author

Reviewer #1: Two important general issues limit the publication of the study and need to be addressed. First, the authors need to better focus the information that this study provides. In fact, it shows the trend of humoral and cellular immunity directed towards a SARS-CoV-2 antigen, namely the spike protein. This information may be of some use in the context of vaccination campaigns with vaccines based on the spike protein. But it cannot be a general biological information on immunity against the virus, because the authors should have evaluated the cellular humoral response against other viral antigens, as carried out in numerous studies (see for example, Le Bert Nature 2020). Therefore, starting with the title and throughout the text, the authors must make it clear that these are humoral and cellular responses against the spike protein and not generically against SARS-CoV-2.

Second, the authors should state whether the test they use is calibrated against the WHO International Standard for anti-SARS-CoV-2 immunoglobulin, a standard that the international community has endeavored to produce in order to promote the standardization of SARS-CoV-2 serological methods and allowing for comparison and harmonization of datasets across laboratories. If the test they use is not calibrated against the WHO International Standard for anti-SARS-CoV-2 immunoglobulin, their data is of little use to the scientific community.

Minor: Authors must provide the code of the diagnostic test used for the determination of IgG / IgA and IGRA.

Reviewer #2: In this paper Rebholz, Braun et al. sought to assess IgG, IgA and SARS-COV-2-specific INF-gamma release in a cohort of patients and compared these data with same data collected in the same area about 8 months before. Main findings include: i) in patients with previous IgG/IgA positivity, IgG levels reduced at the follow up time, whereas IgA remained stable; ii) no correlation was shown between Ig levels and IFN-gamma response.

The authors claim that early T-cell response was associated to mild Covid-19, and yet also discordant data have been reported. The authors should tune their conclusions and discuss their data also in the light of discordant findings (e.g. Tincati et al. Frontiers Immunol 2020).

In the methods section, what do the authors mean by “potentially acutely infected but asymptomatic cases”? A more detailed definition of this group of patients is strongly needed.

I suggest that the title is changed as in its present form is quite generic and does not really focus on the research presented.

6. PLOS authors have the option to publish the peer review history of their article (what does this mean?). If published, this will include your full peer review and any attached files.

Do you want your identity to be public for this peer review? For information about this choice, including consent withdrawal, please see our Privacy Policy.

Reviewer #1: No

Reviewer #2: Yes: Giulia Marchetti

---

## [Author Response · Author response to Decision Letter 0]

22 Jun 2022

Point-to-point reply to the reviewers

We would like to thank both reviewers for their very valuable input, which helped us to markedly improve our manuscript!

@Reviewer 1

“Two important general issues limit the publication of the study and need to be addressed. First, the authors need to better focus the information that this study provides. In fact, it shows the trend of humoral and cellular immunity directed towards a SARS-CoV-2 antigen, namely the spike protein. This information may be of some use in the context of vaccination campaigns with vaccines based on the spike protein. But it cannot be a general biological information on immunity against the virus, because the authors should have evaluated the cellular humoral response against other viral antigens, as carried out in numerous studies (see for example, Le Bert Nature 2020). Therefore, starting with the title and throughout the text, the authors must make it clear that these are humoral and cellular responses against the spike protein and not generically against SARS-CoV-2.”

We agree with reviewer 1 that our tests addressed the humoral and T cell responses of the SARS-CoV-2 spike protein only, and that the data cannot be easily extrapolated to the SARS-CoV-2 virus. As suggested by reviewer 1 we changed the title of the manuscript, and checked the text of the whole manuscript to be more specific here.

“Second, the authors should state whether the test they use is calibrated against the WHO International Standard for anti-SARS-CoV-2 immunoglobulin, a standard that the international community has endeavored to produce in order to promote the standardization of SARS-CoV-2 serological methods and allowing for comparison and harmonization of datasets across laboratories. If the test they use is not calibrated against the WHO International Standard for anti-SARS-CoV-2 immunoglobulin, their data is of little use to the scientific community.”

The serological test used (Euroimmun AG, Lübeck, Germany) was calibrated against the WHO international standard for anti-SARS-CoV-2 immunoglobulin. Tests were performed in a certified diagnostic laboratory (Bioscientia, Ingelheim, Germany), which participates regularly in inter-laboratory performance testing (ring tests), to ensure the highest accuracy and specificity of the applied tests.

@Reviewer 2

“In this paper Rebholz, Braun et al. sought to assess IgG, IgA and SARS-COV-2-specific INF-gamma release in a cohort of patients and compared these data with same data collected in the same area about 8 months before. Main findings include: i) in patients with previous IgG/IgA positivity, IgG levels reduced at the follow up time, whereas IgA remained stable; ii) no correlation was shown between Ig levels and IFN-gamma response.

The authors claim that early T-cell response was associated to mild Covid-19, and yet also discordant data have been reported. The authors should tune their conclusions and discuss their data also in the light of discordant findings (e.g. Tincati et al. Frontiers Immunol 2020).”

Thanks a lot for this valuable hint! We cited the according references, in order to present a more balanced view on the T cell response related to disease severity.

“In the methods section, what do the authors mean by “potentially acutely infected but asymptomatic cases”? A more detailed definition of this group of patients is strongly needed.”

Thanks for the remark, we rephrased the sentence, in order to make it clearer:

The cohort consisted of non-infected participants, and formerly infected but recovered cases. We cannot exclude, that cases were also included with asymptomatic infections at the time point of sample collection.

“I suggest that the title is changed as in its present form is quite generic and does not really focus on the research presented.”

We agree with Reviewer 2 that the title is too generic. Indeed, this was also addressed by Reviewer 1. We changed the title accordingly:

Longitudinal monitoring of SARS-CoV-2 spike protein-specific antibody responses in Lower Austria

---

## [Editor Report · Decision Letter 1]

30 Jun 2022

Longitudinal monitoring of SARS-CoV-2 spike protein-specific anti­body responses in Lower Austria

PONE-D-21-29769R1

Dear Dr. Braun,

We’re pleased to inform you that your manuscript has been judged scientifically suitable for publication and will be formally accepted for publication once it meets all outstanding technical requirements.

Kind regards,

Carlo Torti

Academic Editor

PLOS ONE
---

## [Editor Report · Acceptance letter]

7 Jul 2022

PONE-D-21-29769R1 

Longitudinal monitoring of SARS-CoV-2 spike protein-specific anti­body responses in Lower Austria 

Dear Dr. Braun:

I'm pleased to inform you that your manuscript has been deemed suitable for publication in PLOS ONE. Congratulations! Your manuscript is now with our production department. 

Kind regards, 

on behalf of

Dr. Carlo Torti 

Academic Editor

PLOS ONE